# Seasonal variations in social contact patterns in a rural population in north India: Implications for pandemic control

Sargun Nagpal[1], Rakesh Kumar[2], Riz Fernando Noronha[1], Supriya Kumar[3], Debayan Gupta[1], Ritvik Amarchand[2], Mudita Gosain[2], Hanspria Sharma[2], Gautam I. Menon[1]*, Anand Krishnan[2]*

1 Ashoka University, Sonepat, Haryana, India, 2 All India Institute of Medical Sciences, New Delhi, India, 3 Bill and Melinda Gates Foundation, Seattle, WA, United States of America

* gautam.menon@ashoka.edu.in (GIM); kanand@aiims.ac.in (AK)

Data Availability Statement: All relevant data are within the paper and its Supporting Information files.

## Abstract

Social contact mixing patterns are critical to model the transmission of communicable diseases, and have been employed to model disease outbreaks including COVID-19. Nonetheless, there is a paucity of studies on contact mixing in low and middle-income countries such as India. Furthermore, mathematical models of disease outbreaks do not account for the temporal nature of social contacts. We conducted a longitudinal study of social contacts in rural north India across three seasons and analysed the temporal differences in contact patterns. A contact diary survey was performed across three seasons from October 2015–16, in which participants were queried on the number, duration, and characteristics of contacts that occurred on the previous day. A total of 8,421 responses from 3,052 respondents (49% females) recorded characteristics of 180,073 contacts. Respondents reported a significantly higher number and duration of contacts in the winter, followed by the summer and the monsoon season (Nemenyi post-hoc, p<0.001). Participants aged 0–9 years and 10–19 years of age reported the highest median number of contacts (16 (IQR 12–21), 17 (IQR 13–24) respectively) and were found to have the highest node centrality in the social network of the region (pageranks = 0.20, 0.17). A large proportion (>80%) of contacts that were reported in schools or on public transport involved physical contact. To the best of our knowledge, our study is the first from India to show that contact mixing patterns vary by the time of the year and provides useful implications for pandemic control. We compared the differences in the number, duration and location of contacts by age-group and gender, and studied the impact of the season, age-group, employment and day of the week on the number and duration of contacts using multivariate negative binomial regression. We created a social network to further understand the age and gender-specific contact patterns, and used the contact matrices in each season to parameterise a nine-compartment agent-based model for simulating a COVID-19 epidemic in each season. Our results can be used to parameterize more accurate mathematical models for prediction of epidemiological trends of infections in rural India.

**Funding:** This work was supported by CDC co-operative agreement number 5U01IP000492 with AK and 14IPA141265003 with SK. Apart from the listed authors, the funders had no role in study design, data collection and analysis, decision to publish, or preparation of the manuscript.The findings and conclusions in this report are those of the authors and do not necessarily represent the official position of the Centers for Disease Control and Prevention. There was no additional external funding received for this study.

**Competing interests:** The authors have declared that no competing interests exist.

## Introduction

The study of contact mixing patterns has received considerable attention for its usefulness in parameterizing mathematical models of disease outbreaks and assessing the impact of intervention strategies. Prior to the COVID-19 pandemic, such studies have mostly focused on regions such as Europe [1,2], USA [3], China [4] and Thailand [5]. These studies have analysed the number and duration of contacts in various social settings (home, school, work, other), and whether the contacts were conversational or physical. Further studies [1,6–8] have made use of age and gender stratified contact mixing patterns to simulate infection epidemics and study epidemiological parameters such as disease prevalence, epidemic size, and cumulative infections. More recently, contact tracing and social mixing data from China and Europe have been used to model the transmission of SARS-CoV-2 and understand the effectiveness of measures such as social distancing and school closure [9–12]. However, systematic reviews of contact patterns [13,14] indicate a distinct lack of data from low-income countries.

Furthermore, while these studies are valuable for understanding the overall contact mixing patterns at a geographical location, they do not account for the temporal aspect of social contacts. Mathematical models of disease transmission have been developed with the assumption that social contacts do not change with time. Information on the timing of contacts can aid our understanding of how an infection might progress in different seasons and whether or not the same control strategy might be appropriate at different times of the year. Eames et al. [15] analysed the dynamic contact patterns in the UK, but only compared contacts during school-holidays with term-time. Additionally, different disconnected periods were merged to form a single holiday period. Fournet and Barrat [16] studied the longitudinal changes in social contacts over two years but their study was limited to a sample of high-school students in France. Another study by Béraud et al. [17] looked at the temporal variations in contact patterns in France but their study was limited to a period of four months. Jiang et al. [18] collected contact data from a population in China over three visits across several years, but did not analyse temporal differences in the contact patterns.

Although some studies have attempted to generate synthetic contact matrices (generated using surveys and demographic data) [19,20], there is a paucity of studies collecting and analyzing empirical data (gathered directly from contact diaries) from rural India—the most populous country in the world with a population of over 1.3 billion and 65% rural population [21]. During 2015–2016, we collected contact mixing data from Ballabgarh, a rural town in Haryana, India, and published the results of contact mixing for a single season [22]. In this study, we report the findings of a contact diary survey conducted over three seasons across the year in Ballabgarh. Using this data, we compared the differences in the number, duration and location of contacts by age-group and gender, studying the impact of the season, age-group, employment and day of the week on the number and duration of contacts using multivariate negative binomial regression. We then created a social network to further understand the age and gender specific contact patterns, using these contact matrices in each season to parameterise a stochastic agent-based model for simulating a COVID-19 epidemic in each season.

## Materials & methods

### Field site

Our data were collected in Ballabgarh, located in the state of Haryana, India—an area characterised by rural agrarian communities and multi-generational households. A convenience sample of households consisting of 3052 respondents was taken from five villages, which were under an Acute Respiratory Infection (ARI) surveillance program that included weekly

household visits by trained healthcare professionals to document ARI and influenza episodes among children aged less than 10 years and adults over 60 years of age.

## Ethical review

This study was reviewed and approved by the ethics boards at the All India Institute of Medical Sciences, New Delhi (IEC/NP-121/10-4-2015), as well as at the University of Pittsburgh (PRO15100147) and the Centers for Disease Control and Prevention, Atlanta (FWA 00014191). We received written consent from all participants 18 years of age and above. All those between 7 years and 18 years provided written assent in addition to the written consent of the caregivers. Caregivers provided written consent for participants below 7 years of age.

## Contact diary survey

Respondents completed a structured questionnaire regarding their contacts in the past 24 hours during a face-to-face interview. A caregiver responded on behalf of children under 6 years of age, while children aged 6–10 responded in the presence of a caregiver. An interviewer of the same gender interviewed all respondents aged 11–18 years. A contact was defined as a face-to-face conversation within a distance of three feet, which may or may not have involved physical touch. Respondents were asked about the age and sex of their contacts, the place of contact (home, school, work, transport, or other), and whether the contacts were conversational or physical. In addition, the respondents were provided the option for reporting simultaneous encounters with more than one individual as "group" contacts, including the group size, duration of encounter, and the age range of individuals in the group. Details such as the age and gender of each individual in the group were not recorded. Respondents were interviewed three times in a period of 13 months. The data were gathered over three phases: October 2015—February 2016 (winter); March—June 2016 (summer); and July—October 2016 (monsoon). A small fraction (6.8%) of Phase 1 records were from October 2015, and henceforth we refer to the three phases as seasons.

## Data cleaning

The data contained missing values and typographical errors which had to be accounted for by cleaning the dataset. We first sanitised the dates and calculated respondent ages from their date of birth and the date of interview to create a minimally-cleaned dataset. We further imputed contact genders by checking instances of the same name; the number, duration, and age range of group contacts by using the median of group contacts at the same location; and the contact ages by sampling from age distributions of similar respondents. This comprised a fully cleaned dataset. S1 Table contains a detailed description of the attributes cleaned and the methodology followed for imputation or correction. Our results were consistent with both datasets, and all results displayed in the manuscript are based on the fully cleaned dataset.

## Comparison of contacts within and across the seasons

We performed two types of analyses of the data, the first being an analysis within each season, similar to the work of Kumar et al. [22], where we identified the differences in contact patterns of age groups for each season individually. For this study, statistical methods for independent samples were used for comparison (Kruskal Wallis, Dunn's test) since each age category had distinct individuals, and these samples could be assumed independent. Further, we analysed the differences in contact patterns across the three seasons. For this study, participants who responded in two or more seasons were considered for analysis and statistical methods for

dependent samples were used (Friedman, Nemenyi test) since the contacts of the same individual across multiple seasons cannot be assumed to be independent.

## Number of contacts

For each respondent, we report the number of individual contacts over the previous day, the number of people contacted in a group setting and the total number of contacts by adding the individual and group contacts.

We defined "superspreaders" as respondents having >95th percentile contacts in each season. Since these respondents met a large number of people, they could be potential superspreaders of a communicable disease.

## Duration of contacts

The reported durations of contacts were recorded on a categorical scale: "<5 minutes", "5–14 minutes", "15–59 minutes", "1–4 hours" and ">4 hours". These were converted to numerical values by computing the mean duration for each interval. For instance, for the 15–59 minutes interval, a contact duration of 37.5 minutes was used. For the >4h category, we set the upper limit as 8 hours, the usual maximum for a working day. The total duration of group contact for each respondent was calculated by multiplying the group size with the duration of contact with the group. The total time spent in contact for each respondent was calculated as the sum of individual and group contact durations and reported as person-hours.

## Age and gender-stratified contacts

We calculated contact numbers and durations stratified by the respondent age-groups (0–9, 10–19, 20–29, 30–39, 40–49, 50–59, 60–69 and 70+). For analyses comparing respondents within a season, we used the Kruskal-Wallis nonparametric H test [23] to identify differences in the contact rates, followed by Dunn's test to examine pairwise differences between age-categories, with the Bonferroni correction to account for multiple comparisons. For comparisons across the three seasons, we selected respondents who were present in more than one season and used statistical tests for dependent samples, namely the Friedman non-parametric test, followed by a Nemenyi post-hoc test.

Further, we plotted gender stratified boxplots for each age category and analysed if a significant difference existed between the number of contacts for males and females for each age group using the Kruskal-Wallis test. The effect size was calculated to quantify the difference in total contacts using the Cohen's d [24] metric.

$$d = \frac{\bar{x_1} - \bar{x_2}}{s},$$

where $\bar{x_1}$ and $\bar{x_2}$ are the mean contacts for males and females respectively and S is the pooled standard deviation, given by:

$$s = \sqrt{((n_1 - 1)s_1^2 + (n_2 - 1)s_2^2)/(n_1 + n_2 - 2)}$$

*where $n_1$, $n_2$ are the number of males and females, and $s_1$, $s_2$ are the standard deviations of the number of contacts of males and females respectively.*

## Social networks to visualise contact patterns

We created social networks to visualise the contact mixing patterns at Ballabgarh. Since our survey was conducted on a sample of the Ballabgarh population, and all reported contacts

were not respondents in the survey, we represented contacts using a directed graph. In our networks, the nodes represent the ten year age categories with their genders and the node sizes represent the total median contacts. The weighted directed edge from a node A to B represents the median number of contacts A had with B. The visualisation was created with Gephi [25] and the graph layout was generated using the ForceAtlas2 algorithm [26], taking into account the node sizes. The PageRank algorithm [27] was used to calculate the node centrality of each node and nodes were colored based on their PageRank, with darker colours representing higher values.

## Contact setting

To understand if the contact patterns differed inside and outside home, boxplots for individual, group, and total number and duration of contacts were plotted for each season. Wilcoxon signed rank test [28] was used to check if the difference between inside and outside home contacts was significant.

The percentage of individual contacts involving a physical touch were calculated at each location (home, school, work, transport, other) for every respondent. Barplots indicating the mean values and 95% confidence intervals were plotted for the three seasons. To analyse if the proportions of physical contacts differed across the seasons, respondents who responded in more than one season were filtered and the Friedman and Nemenyi post-hoc test were used to examine pairwise differences between the seasons. Barplots for the proportion of physical contacts with the contact duration and contact frequency for each season were also plotted.

## Group contact purposes

Every respondent specified the purpose of each of their group contacts in a free-write format. These strings were inconsistent and often had spelling mistakes. To understand the difference in group contacts across the three seasons, we created 13 contact reason categories such as school, politics, weddings, work, and worship. A set of keywords was defined for each category (S2 Table). Keyword matching was performed to map the contact reason strings to these categories. One string could map to multiple categories and a contact belonging to none of the other categories was mapped to the 'other' category. Barplots were created to highlight the differences in the distributions of group contacts for each category across the three seasons. Both the total size of group contact in each category and the number of people who reported a group contact of each category were visualised.

## Age-assortative mixing matrices

We calculated age-assortative mixing matrices for both the number and duration of contacts. These matrices represent the average number or duration of contacts between each pair of age categories. For group contacts, the individual age of each member in the group was not known, and only the least and highest age in the group was present in the dataset. Therefore, we sampled ages between the lower and upper age based on the census age proportions of rural Faridabad. These proportions were computed using the age-wise population counts for rural Faridabad, Haryana obtained from the 2011 census of India [29].

## Observed to expected age-assortative mixing matrices

We calculated the Observed to Expected (O/E) mixing matrices to evaluate whether the actual number or duration of contacts between a pair of groups was higher or lower than the number we would expect if mixing were proportional to the population sizes. In a conventional mixing

matrix, a high value between two groups could be obtained just because of a high number of individuals in one of the age categories. Therefore, calculating the O/E matrices helps us to account for the population sizes. A value greater than 1 in the matrix signifies a greater than expected mixing between two groups while a value less than 1 signifies the opposite.

To calculate the O/E matrix, 1000 bootstrap samples were drawn from the contact mixing data. The observed number and duration of contacts were calculated for each sample. The expected contacts of a respondent category with other contact categories were calculated by multiplying the total contacts for the respondent category with the census population proportions of the contact categories. We present these observed by expected matrices with 95% confidence intervals for both the number and duration of contacts, stratified by season, as well as by respondent gender.

The Q-index [30] was used as a measure of the assortativity of the mixing matrices. It ranges from 0 to 1, with 0 representing random mixing and 1 representing perfect assortativity.

$$Q = \frac{Tr(P) - 1}{n - 1}$$

where $P$ is the contact matrix normalised to a left-stochastic matrix and $n$ is the number of age categories.

## Regression models to predict contacts and understand the effect of explanatory variables

We fitted univariate regression models stratified by gender, with the age as the independent variable and the median number of contacts as the dependent variable. To avoid outliers, the median contacts for an age were only calculated if there were a minimum of three respondents of that age. A polynomial regression model of degree 5 was fitted on the data, based on evaluation metrics like the Mean Squared Error and Mean Absolute Error. This model captured the complexity of the data, while still not being over-parameterized.

We also constructed a multivariate negative binomial regression model to understand the independent effect of the season, age-groups (in ten-year categories), employment, and whether the day was a weekend (while controlling for other covariates) on the number and duration of contacts. Respondents with occupations listed as unemployed, retired, dependents, aged individuals, housewives, and girls doing household chores were treated as unemployed for the regression model. The age category of 10–19 years old (the group with the highest contacts in previous studies [1]) and the winter season were chosen as reference categories. The 'NegativeBinomial' GLM family was used from the 'statsmodels' package [31] in Python with the default hyperparameter values. We present the adjusted rate ratios for the model, along with their 95% confidence intervals.

## Generating a synthetic population that mimics contact patterns

In order to simulate an epidemic in the population, we generated a synthetic population with contact patterns similar to that of our study. We generated a population of size 10,000 (one for each season) with an age distribution similar to that of the census data for Faridabad, and used a Monte-Carlo approach following the simulated annealing metaheuristic to assign every agent a house and a workplace, such that the age-stratified contact pattern in homes and workplaces resembled the observed pattern of contacts within the home and outside the home respectively.

## Agent-based modelling to simulate infection spread using contact mixing data

We used the synthetic populations to perform epidemiological simulations using BharatSim [32], an agent-based simulation engine. In an agent-based model, we initialise several 'agents' each with their own schedule, household and workplace, and simulate their interactions with one another. The rates and parameters used were derived from Kerr et al. [33] and are presented in S4 and S5 Tables. We simulated the spread of COVID-19 (as a model for a respiratory disease) using the compartmental model described by Hazra et al. [34] (S10 Fig) in the population in order to track differences in the spread of the infection due to the seasons.

## Results

A total of 8,421 responses were obtained from 3,052 respondents across the three seasons. Table 1 shows the characteristics of those who participated in the survey. S1 Fig illustrates the season-wise demographic details along with counts for new enrollments and loss to follow-up. A total of over 120,000 individual contacts and 58,000 group contacts were reported, for a total of 180,000 contacts. 2,913 respondents (95.4%) responded in more than one survey and were considered for a longitudinal analysis across the seasons.

### Number of contacts

The total number of contacts differed across the seasons (Friedman $\chi 2$ = 213.60, p-value $< 0.001$). Respondents had the highest median contacts in winter, while the least contacts in monsoon (Nemenyi post-hoc test, $p < 0.05$). Table 2 shows the median and total number of individual, group and total contacts reported in each season. The winter season had a significantly higher median group contacts compared to the other seasons. Furthermore, the number of contacts differed by age group in each season (Kruskal-Wallis, $p < 0.05$) (Fig 1A–1C). Respondents aged 10–19 years had the highest median contacts and respondents aged 70 and over had the least median contacts in each season but were not significantly different from all other age categories (Fig 1D–1F). For all age categories, the median number of contacts decreased monotonically from the winter to the monsoon season (Fig 1G).

**Table 1. Characteristics of survey participants, stratified by gender.** The numbers indicate the number of responses obtained, except for the 'Total' row in the 'n' column, which represents the total number of unique respondents who answered the survey.

| Characteristic | | n | Males n (%) | Females n (%) |
|---|---|---|---|---|
| **Total** | | 3052 | 4278 (50.7) | 4153 (49.3) |
| Age | 0–9 | 814 | 449 (55.2) | 365 (44.8) |
| | 10–19 | 583 | 289 (49.6) | 294 (50.4) |
| | 20–29 | 532 | 245 (46.1) | 287 (53.9) |
| | 30–39 | 465 | 250 (53.8) | 215 (46.2) |
| | 40–49 | 177 | 104 (58.8) | 73 (41.2) |
| | 50–59 | 85 | 34 (40.0) | 51 (60.0) |
| | 60–69 | 276 | 126 (45.7) | 150 (54.3) |
| | 70+ | 120 | 57 (47.5) | 63 (52.5) |
| Season | Winter (Oct-Feb) | 2942 | 1504 (51.1) | 1438 (48.9) |
| | Summer (Mar-Jun) | 2627 | 1328 (50.6) | 1299 (49.4) |
| | Monsoon (Jul-Oct) | 2852 | 1446 (50.7) | 1406 (49.3) |
| Employment | Employed | 722 | 644 (89.2) | 78 (10.8) |
| | Unemployed | 2330 | 910 (39.1) | 1420 (60.9) |

**Table 2. Median and cumulative number of individual, group and total contacts reported in each season.** The interquartile range (IQR) for the total number of contacts is also indicated. Group Contacts represent the number of people met in group settings, Total contacts represents the sum of individual and group contacts.

| | Individual Contacts | | Group Contacts | | Total Contacts | | |
|---|---|---|---|---|---|---|---|
| | Median | Cumulative | Median | Cumulative | Median | Cumulative | IQR |
| Winter | 15 | 46,100 | 35 | 30,404 | 17 | 76,504 | 12–26 |
| Summer | 13 | 37,273 | 20 | 13,809 | 15 | 51,082 | 11–22 |
| Monsoon | 13 | 38,682 | 25 | 13,795 | 14 | 52,477 | 10–20 |

The threshold number of contacts needed to be considered a superspreader (95th percentile) differed by season, from 72 in winter to 48 in both summer and monsoon. Only 11 participants (3.7% of superspreaders) were common across the three seasons, which indicates that a very small proportion of superspreaders in one season were superspreaders in other seasons.

## Number of contacts by gender

In every season, women had a significantly higher number of contacts at home and a lower number of contacts outside home (S7 Fig). The 0–9 and >49 age categories had a similar number of contacts across gender (Table 3 and Fig 2A), whereas a significant difference existed between males and females in the middle-aged categories. The 30–39 age category had the highest effect size in each season. This difference could be due to employment—a higher proportion of adult male respondents were employed outside of the home compared to their female counterparts, and employed individuals were found to have more contacts than unemployed ones (S8 Fig).

For males, the number of contacts significantly differed by season (Friedman Test, $p < 0.05$) for all age groups except the 50–59 years age category (Fig 2B). Males were found to have a higher median number of contacts in winter for all but the 60+ age groups (Nemenyi post-hoc test, $p < 0.05$). For females, the number of contacts significantly differed by season for all but the 40–49 age category (Fig 2C). Females had the lowest median for almost all age groups in monsoon, however a significant difference did not exist between winter and summer for most age categories.

The individual social contact network of Ballabgarh (Fig 2D) shows that both males and females < 20 years of age had high numbers of contacts and occupy a central position in the social network. Males in age category 0–9 years had the highest pagerank, followed by their female counterparts. Adults over 40 years of age appeared at the boundary of the network. 20–29 and 30–39 males had higher median number of contacts and were more centrally located than their female counterparts. A clustering of female and male nodes can be observed in the left and right halves of the network, which suggests a gender assortativity in contacts.

## Contact setting

We stratified the contacts by whether they occurred within or outside the respondents' homes. Both the numbers and durations of group and total contacts were found to be significantly different between the two locations in each season (Wilcoxon signed rank test, p-value < 0.05). The median number of group contacts were higher outside home, while the total contacts were higher within homes (Fig 3A and 3B). Individual contacts were found to have the same trends as total contacts. Group contacts in winter had a higher spread than the other seasons (Fig 3A) which was caused by a large number of wedding, shopping, politics and school related contacts being reported at home. Contact durations follow a similar pattern (Fig 3C and 3D),

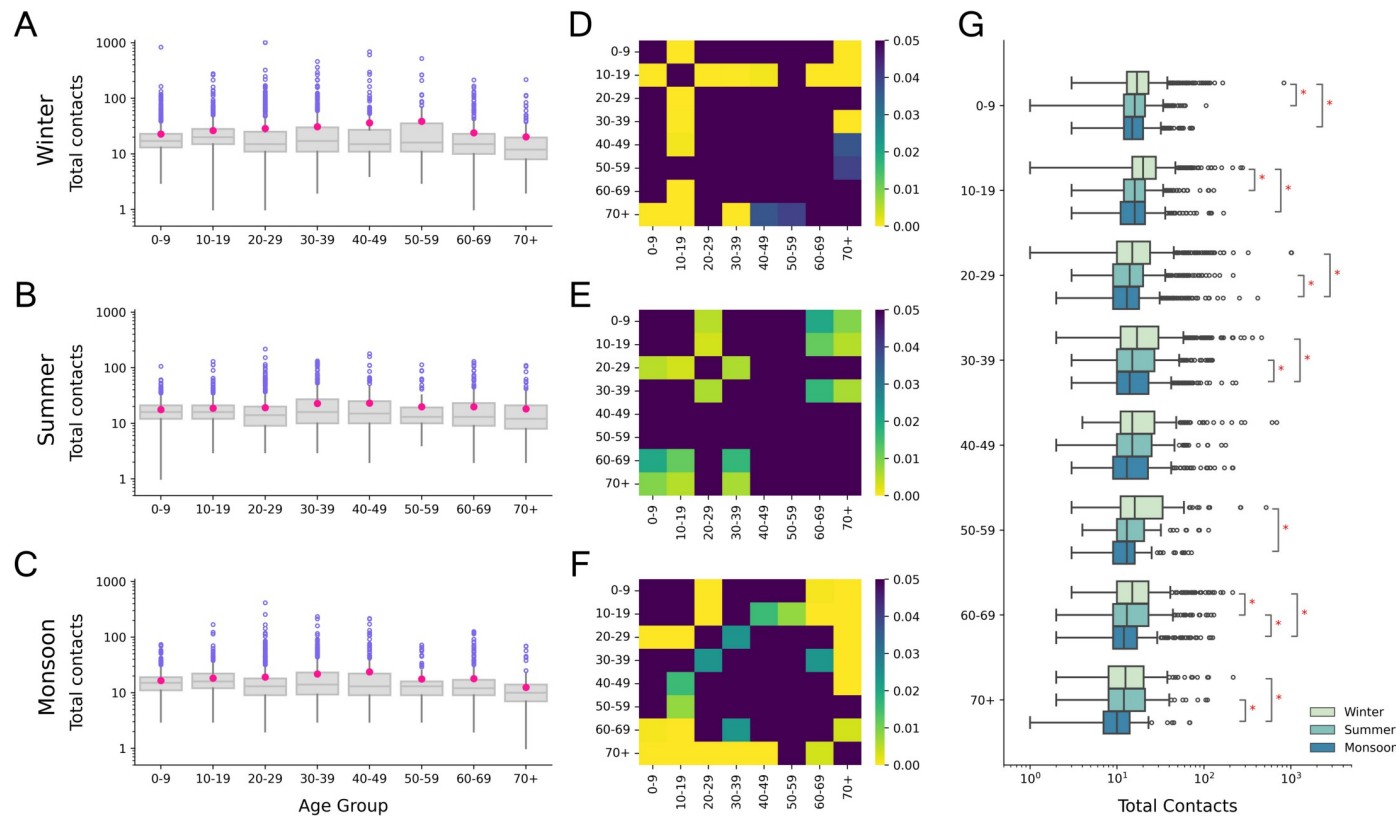

**Fig 1. Age stratified comparison of total contacts reported in each season.** (A), (B), (C) Age stratified number of contacts in winter, summer and monsoon respectively. Pink dots in the boxplots represent the mean values. (D), (E), (F) Heatmaps showing p-values from Dunn's post-hoc test for winter, summer and monsoon respectively. Dark regions represent pairs of age categories for which the difference in the number of contacts is not significant (p>0.05). (G) Comparison of number of contacts for age categories across the three seasons. Pairs of age categories with a significant difference in contacts (Nemenyi post-hoc test, p<0.05) are marked with an asterisk symbol.

except monsoon, which had a higher median duration of group contacts within home (Fig 3C). This may be due to people spending more time at their homes in the monsoon season.

A longitudinal study of each individual across the seasons revealed that the proportion of individual contacts that involved a physical touch were significantly different across the three seasons (Friedman $\chi^2$ = 78.14, p-value < 0.001). The monsoon season had the highest

**Table 3. Effect size (Cohen's d) for difference in the number of contacts of males and females, stratified by age group.** Bold values represent a significant difference in contacts between males and females (Kruskal Wallis, p-value < 0.05).

| Age Category | Effect size between Males and Females | | |
|---|---|---|---|
| | Winter | Summer | Monsoon |
| 0–9 | -0.050 | 0.047 | 0.017 |
| 10–19 | **0.379** | **0.327** | **0.277** |
| 20–29 | **0.395** | **0.516** | **0.527** |
| 30–39 | **0.501** | **0.738** | **0.550** |
| 40–49 | **0.365** | **0.558** | **0.500** |
| 50–59 | 0.501 | 0.468 | **0.761** |
| 60–69 | 0.202 | **0.417** | 0.170 |
| 70+ | -0.051 | 0.292 | 0.033 |

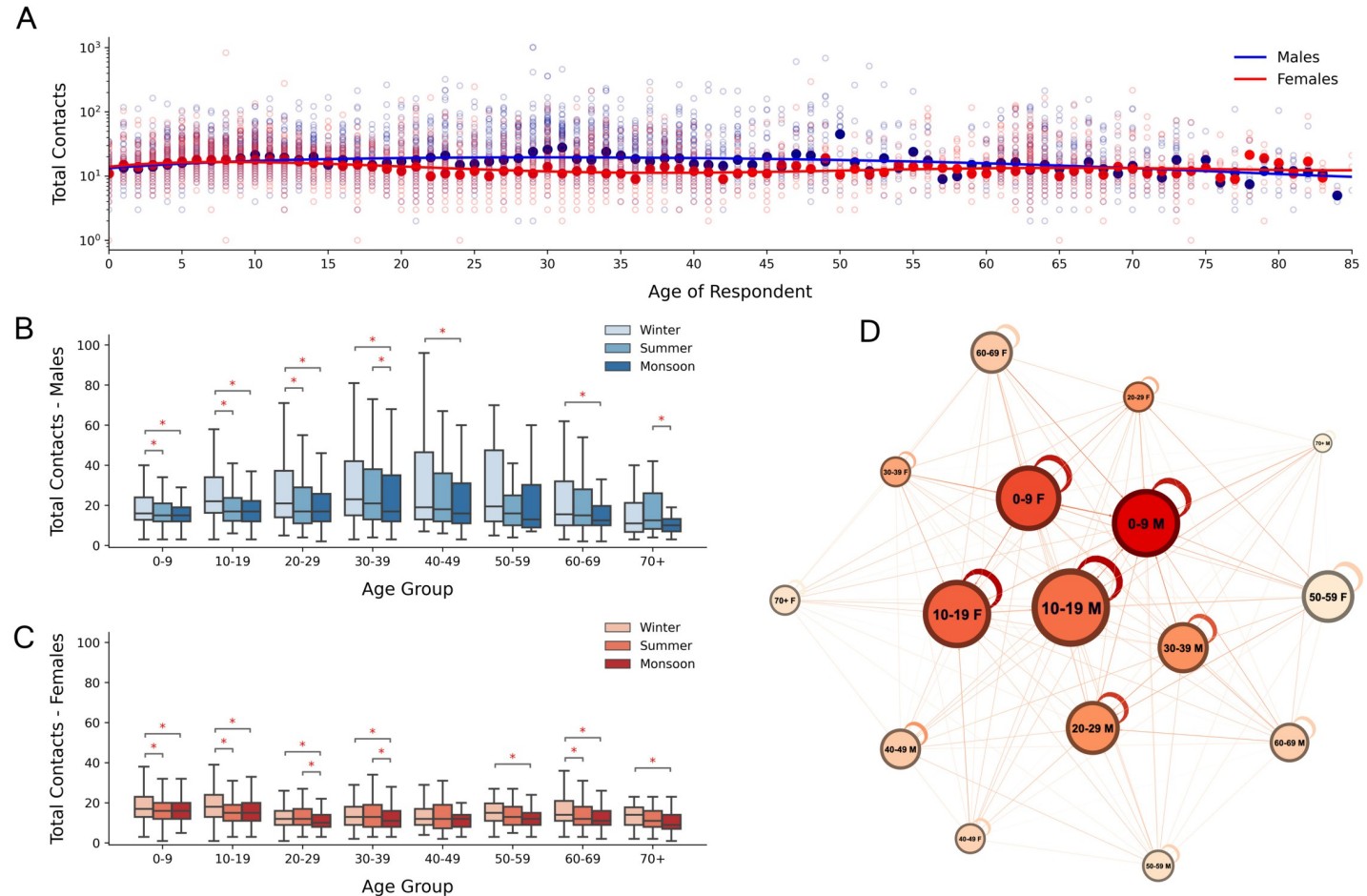

**Fig 2. Gender and age stratified number of contacts.** (A) Scatterplot of number of contacts by age, red representing females and blue representing males. Dark circles indicate median values. Polynomial regression curves are fit to the medians and presented. (B), (C) Boxplots of season and age-stratified number of contacts reported for males and females respectively. Age groups with significant differences across seasons are highlighted with asterisks. (D) Directed graph representing the median number of individual contacts an age category has with every other age category. Higher numbers are depicted with stronger edge weights. Node sizes represent the total median contacts and node colours represent the PageRank of each node (darker = more important).

proportion of physical contacts, while the winter season had the least proportion (Nemenyi post-hoc test, p < 0.05). The same trend was seen at almost all locations (Fig 3E). Contacts at schools and transport involved a very high proportion of physical contacts. We observed an increase in the proportion of contacts that involved physical touch as the duration of contacts increased (S6A Fig). Contacts that occurred with a person who was met often had a higher chance of being physical than those who they had met for the first time (S6B Fig).

## Duration of contacts

As with the number of contacts, the total duration of contacts differed across the seasons (Friedman χ2 = 44.32, p-value = 2.38 × 10^{-10}). Respondents had the highest median duration in winter, while the least duration of contacts in monsoon (Nemenyi post-hoc test, p < 0.05). Winter had a significantly higher median duration of group contacts compared to the other seasons, which was due to a higher number of group contacts as well as a disproportionately large number of contacts with the maximum duration (>4hrs) being reported (S2 Fig). Furthermore, the duration of contacts differed by age group in each season (Kruskal-Wallis,

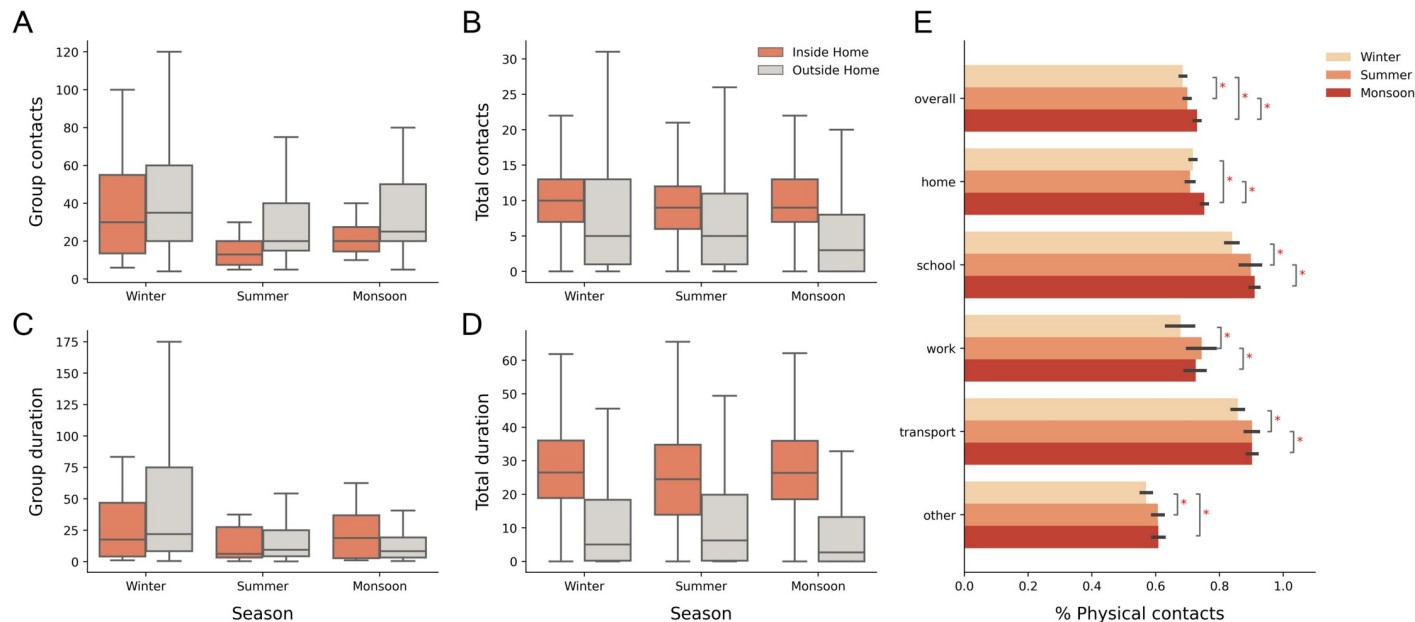

**Fig 3. Contact setting and proportions of physical contacts.** (A), (B) The number of group and total contacts reported inside and outside the home, for each season. (C), (D) The durations (in person-hours) of group and total contacts reported inside and outside the home, for each season. (E) The mean proportion of physical contacts across all respondents observed at every location for each season. Black horizontal lines represent 95% confidence intervals. Significant differences in proportions of physical contacts at each location are represented with asterisks.

$p < 0.05$) (Fig 4A–4C). Young individuals under 20 years of age reported the highest median duration of contacts in all three seasons, significantly different from all other age categories. Old-aged individuals (>70 years) reported the least median duration of contact in all the seasons, significantly different from all age categories <40 years (Fig 4D–4F). Across the seasons, a significant difference was only observed for the 10–19 and >60 years age categories. In both cases, respondents had the highest duration of contacts in winter (Fig 4G).

## Contact purposes

To investigate the differences in group contacts across the seasons, we categorised each group contact into different reasons as described in the Methods section. A large percentage of group contacts in winter were associated with politics (Fig 5A and 5B), due to elections and associated campaigning during that time in Ballabgarh. Upon removing politics-related contacts from winter, the difference in the total contacts across the seasons was still significantly different (Friedman $\chi2 = 204.89$, p-value < 0.001); however, the effect size of winter from summer and monsoon reduced from 0.203 and 0.227 to 0.190 and 0.216 respectively. Therefore, politics related contacts alone cannot account for the difference in contacts and other seasonal factors are responsible for these differences. For instance, a higher proportion of group sizes in winter comprised contacts at schools, madrasas and weddings, but a lower proportion at work (Fig 5A). S3 Table shows a breakdown of group contact reasons by month, in which we find a large number of wedding related contacts in November and February (winter), which are accompanied by high shopping related contacts in the accompanying months. High numbers of school contacts were observed in October and November, and work contacts in February, June and October.

Age-stratified contact reasons (Fig 5C and 5D) give an insight into the possible locations to target if social contacts of an age category need to be curbed. The 0–9 and 10–19 age categories

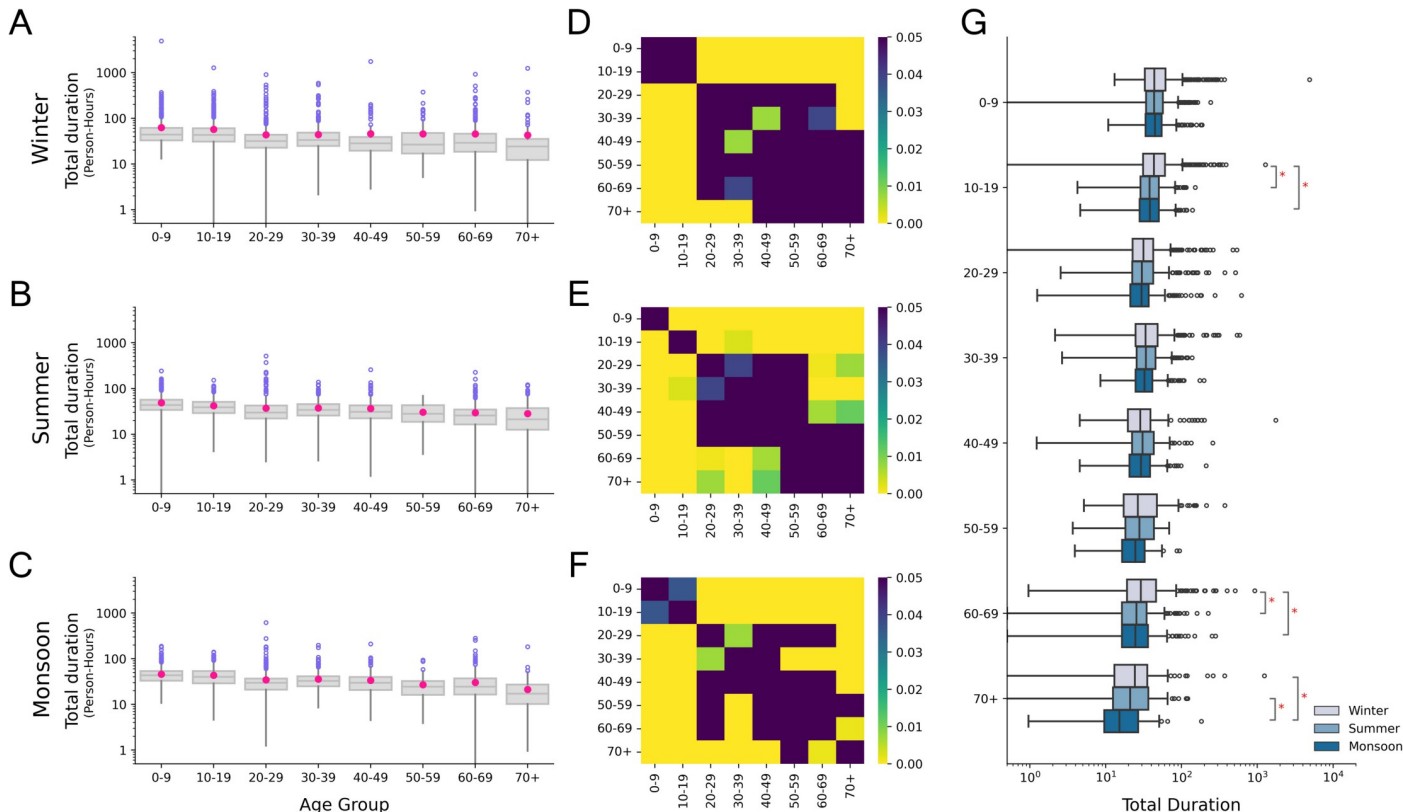

**Fig 4. Total durations (person-hours) of contacts reported by respondents in each season.** (A), (B), (C) Age stratified durations of contacts in winter, summer and monsoon respectively. Pink dots in the boxplots represent the mean values. (D), (E), (F) Heatmaps showing p-values from Dunn's post-hoc test for winter, summer and monsoon respectively. Dark regions represent pairs of age categories for which the difference in the durations of contacts is not significant (p>0.05). (G) Comparison of durations of contacts for age categories across the three seasons. Pairs of age categories with a significant difference (Nemenyi post-hoc test, p<0.05) are marked with an asterisk symbol.

are highlighted as they had a high number and duration of contacts in all the seasons. As expected, a lot of their contacts happened for playing (game) and studying (madrasa, school). Further, it was observed that 0–9 year olds had a high number of contacts in weddings and worship settings, compared to the other age categories.

## Age-assortativity in contacts

We computed the ratio of observed number and durations of contacts to the expected values if mixing were random. Both the number and duration of contacts were found to be age assortative in each season, with a greater than expected mixing with contacts of the same age group (Fig 6). Children (0–9 years) were found to have a greater than expected number of contacts with 60–69 years. Males aged 20–39 had a higher than expected duration of contacts with children (0–9) but a lower than expected number of contacts in all three seasons, which could be due to contacts at home between parents and children. On the other hand, respondents aged 30–59 had a higher than expected number of contacts with 20–29 year olds, but a lower than expected duration of contacts in all three seasons. The average number and duration of contacts between each pair of age categories is presented in S3 Fig.

Contacts in summer were found to be less assortative (lower Q-index) than winter and monsoon. We also generated observed to expected matrices stratified by respondent gender (with the number of contacts in S4 Fig and the duration of contacts in S5 Fig). Male contacts

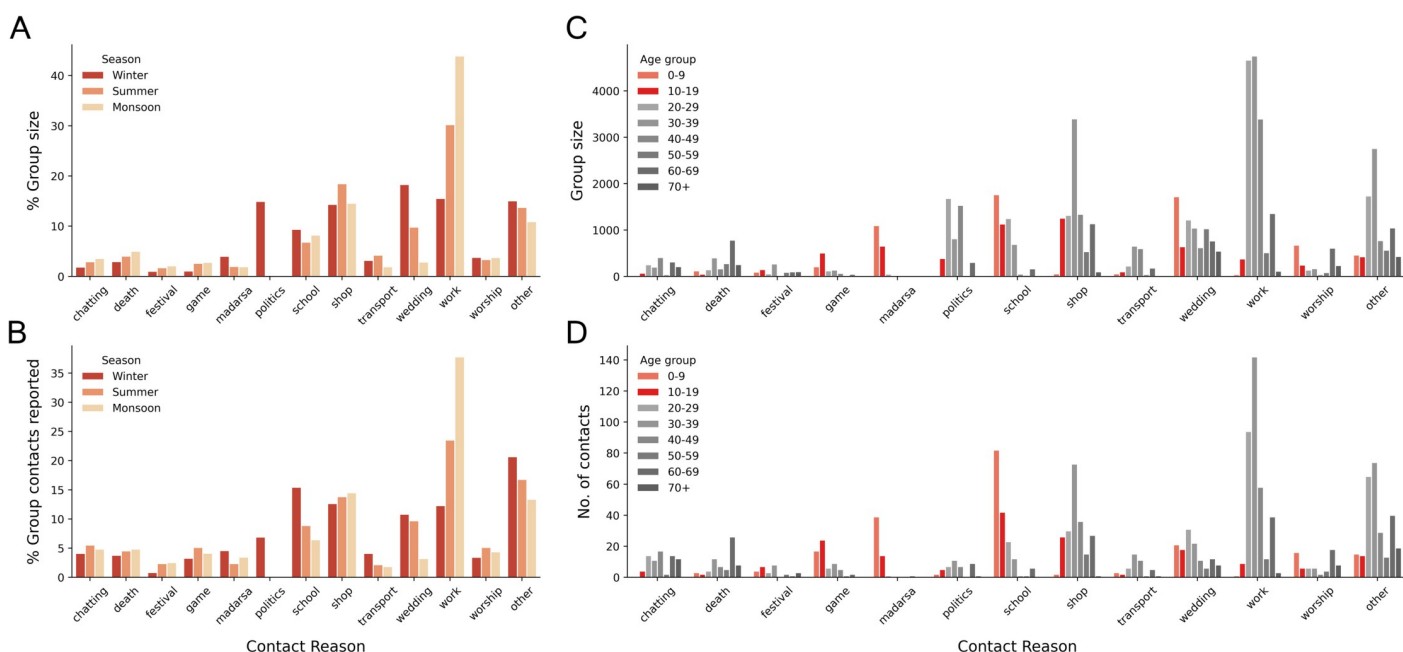

**Fig 5. Season and age stratified group contact purposes.** (A), (B) Season-wise percentages for group contact reasons for the number of people met in a group setting and the number of group contacts reported respectively. (C), (D) Age stratified counts for the number of people met in a group setting and the number of group contacts reported respectively. The 0–9 and 10–19 years categories are highlighted in orange and red respectively.

were found to be more assortative than female contacts, which could be linked to employment outside the home.

## Independent correlates of contacts

As described in the methods, we used a multivariate negative binomial regression model to predict the influence of several factors on the number and duration of contacts. The rate ratios are presented in Table 4.

In terms of both the number and duration of contacts, summer and monsoon had significantly lower contacts compared to winter for both males and females. The intercept values represent the numbers and duration of contacts for males and females in the reference categories. Males in the reference category were found to have a higher number of contacts compared to females, but a similar value for the duration of contact. Males aged 20–49 reported a higher number of contacts than the reference 10–19 category, while the opposite was true for females. Although males aged 20–59 report a higher number of contacts than the reference, they have shorter total contact durations. Employed individuals were seen to have more contacts than their unemployed counterparts. The rate ratios for employed females was higher than those for employed males. The rate ratio was not significant for whether the contact happened on a weekend or weekday.

## Simulation of COVID-19 outbreak in each season

The averaged results from 50 simulations monitoring the spread of COVID-19 in each season can be found in S11 Fig. In general, we observed minor differences in the SIR curves between the three seasons, with the disease spreading marginally faster in the winter compared to the summer and monsoon seasons. This aligns with the general trend of the number of contacts being maximum in winter, followed by summer and monsoon respectively. We stratified the

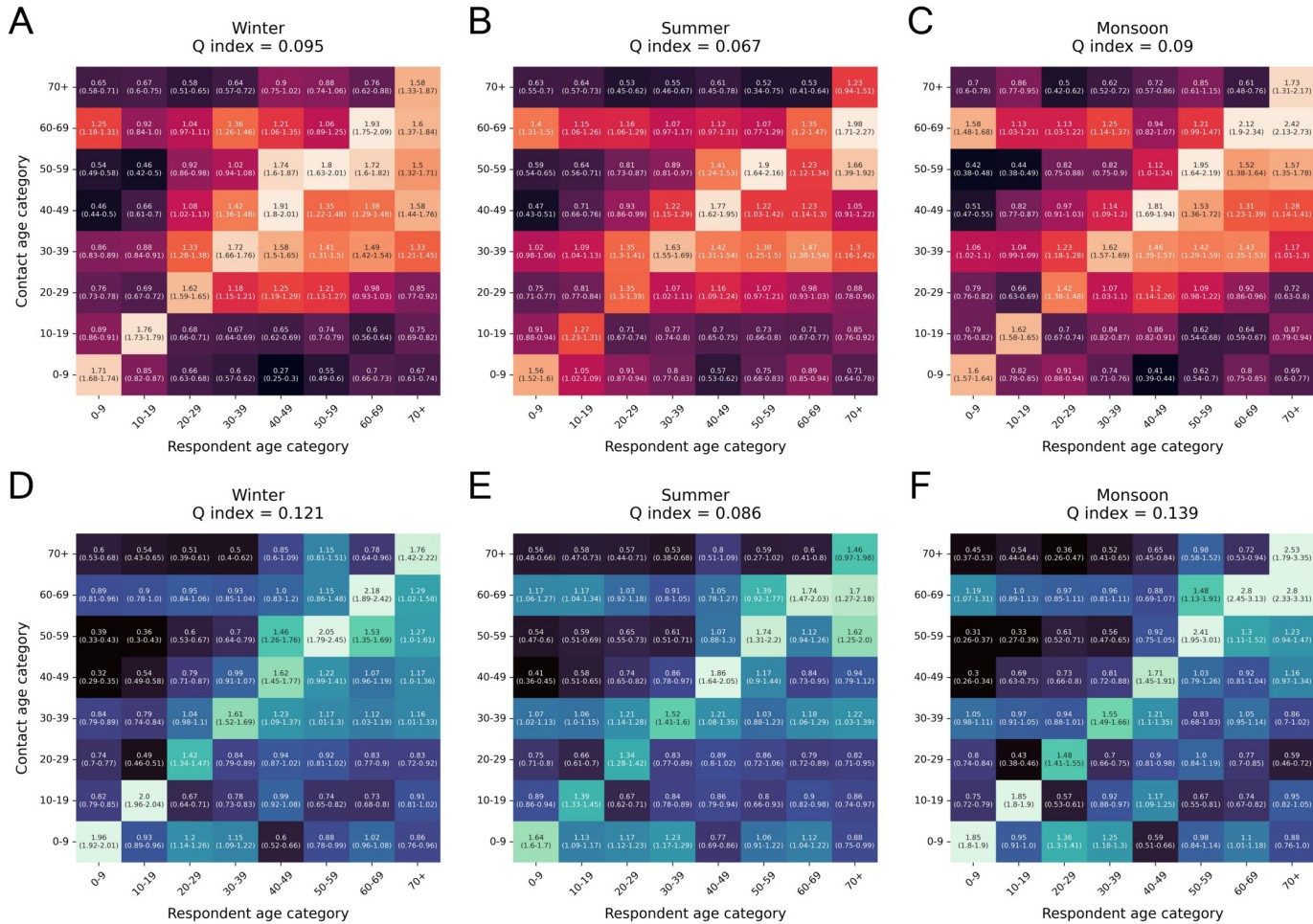

**Fig 6. Observed to expected age assortative mixing matrices for the number and duration of contacts in each season.** (A), (B), (C) Mixing matrices for the total number of contacts in winter, summer and monsoon respectively. Each cell contains the ratio of the observed number of contacts reported by the respondent age-category with the respective contact age category to the expected number of contacts if mixing were proportional to census population proportions in each age category. 95% confidence intervals calculated from 1000 bootstrap samples are also presented. (D), (E), (F) Mixing matrices for the total duration of contacts in winter, summer and monsoon respectively. The Q-index for the degree of assortativity is indicated in the title of each figure.

results by age (S12 Fig), and observed a statistically significant difference in the day that the infection peaks for the 50–59 age category (Kruskal-Wallis $\chi2 = 9.009$, p = 0.011), with the winter season peaking the earliest, followed by summer and subsequently monsoon season. However, a significant difference did not exist for the other age categories. The season-wise variance between individual simulation results can be seen in S13 Fig.

## Discussion

We compared the differences in contact patterns across three seasons in Ballabgarh, a rural region in India. We found that the respondents had a significantly higher number and duration of contacts in the winter season, followed by the summer and monsoon seasons. The same trend was observed for each age category. This is in contrast to a study in France [17], where no clear temporal variation was found across the three time periods. The temporal changes we observed could be attributed to changes in the weather conditions as found in earlier studies [35,36]. The general contact patterns did not appear to change significantly across

**Table 4. Rate ratios (RRs) for factors influencing the number and duration of contacts, stratified by gender.** Bold values indicate statistically significant rate ratios (p-value < 0.05) from a negative binomial generalised linear model. 95% confidence intervals for each RR are given. A RR greater than 1 indicates a greater effect on the number/duration of contacts compared to the reference. The intercept values represent the number or duration of contacts for the reference category.

| Characteristic | | No of contacts Adjusted RR (95% CI) | | Duration of contacts Adjusted RR (95% CI) | |
| --- | --- | --- | --- | --- | --- |
| | | Males | Females | Males | Females |
| Age | 0–9 | **0.8 (0.73–0.87)** | 1.04 (0.94–1.14) | 1.03 (0.94–1.12) | **1.19 (1.09–1.31)** |
| | 10–19 | Reference | Reference | Reference | Reference |
| | 20–29 | **1.17 (1.03–1.32)** | **0.79 (0.72–0.88)** | **0.80 (0.71–0.91)** | **0.76 (0.69–0.84)** |
| | 30–39 | **1.20 (1.03–1.38)** | **0.85 (0.76–0.95)** | **0.78 (0.67–0.90)** | **0.79 (0.71–0.88)** |
| | 40.49 | **1.25 (1.05–1.47)** | **0.85 (0.72–1.00)** | **0.68 (0.58–0.81)** | 0.86 (0.74–1.01) |
| | 50–59 | **1.29 (1.03–1.63)** | 0.93 (0.77–1.12) | **0.65 (0.51–0.82)** | **0.72 (0.60–0.87)** |
| | 60–69 | 0.89 (0.77–1.03) | 0.96 (0.85–1.09) | **0.53 (0.46–0.61)** | **0.89 (0.79–1.00)** |
| | 70+ | **0.72 (0.61–0.86)** | 0.86 (0.73–1.01) | **0.43 (0.36–0.51)** | **0.81 (0.69–0.95)** |
| Season | Winter | Reference | Reference | Reference | Reference |
| | Summer | **0.72 (0.67–0.78)** | **0.78 (0.73–0.85)** | **0.82 (0.76–0.89)** | **0.73 (0.68–0.79)** |
| | Monsoon | **0.67 (0.63–0.72)** | **0.73 (0.68–0.79)** | **0.77 (0.72–0.83)** | **0.71 (0.66–0.76)** |
| Employed | | **1.18 (1.05–1.31)** | **1.40 (1.20–1.62)** | 1.09 (0.98–1.22) | **1.21 (1.04–1.40)** |
| Weekend | | **1.08 (1.01–1.15)** | 1.06 (0.98–1.14) | 1.05 (0.98–1.12) | 1.07 (1.00–1.15) |
| Intercept | | **29.10 (26.76–31.66)** | **21.60 (19.85–23.52)** | **55.77 (51.32–60.64)** | **55.59 (51.11–60.46)** |

seasons, as shown by the degree of similarity of the age-assortative mixing matrices and the results of the agent-based simulation which is in agreement with Fournet and Barrat [16]. The proportion of individual contacts that were physical were found to follow the reverse trend, with the monsoon season having the highest percentage, and the summer season the least. Women had a higher number of contacts inside home, while men had a higher number outside their homes. Social networks are often used to investigate disease spread in a population [37,38]. The PageRank algorithm has been widely used for calculating node importance in social networks, including those for communicable diseases [39,40]. The 0–9 and 10–19 year olds had the highest pagerank in the social network of Ballabgarh. Overall, respondents reported higher median contacts within homes. A large number of group contacts in winters were in weddings and political settings, while in monsoon, the majority group contacts were at work and shops. While superspreading events can be held at any time of the year, the risk of disease spread is larger in seasons with more events of large sizes. Therefore, our analysis suggests that different intervention strategies may be appropriate at different times of the year. The median number of contacts reported overall was 15. In each season, most respondents reported less than 50 total contacts and vanishingly few individuals reported a very high number of contacts (S14 Fig). The steepest decline in the frequency of individuals having a certain number of contacts was found to be in the winter season. Superspreading individuals were also found to differ by season, with only 3.7% of them being superspreaders in all three seasons.

Based on our data, 58% males and 95% females were found to be unemployed. Although the median number of contacts of employed individuals was only slightly higher than unemployed individuals, the contact distribution of employed people had a heavier tail, indicating that some employed people mixed with a very high number of individuals. Furthermore, on stratifying respondents by gender, the median number of contacts of males—both employed (18) and unemployed (16) were found to be higher than their female counterparts (15 and 14 respectively), which implies that employment may not be a confounding variable that alone explains the difference in contacts of males and females.

Prior to correcting for census proportions, we found that the age assortativity of the number of contacts was similar to other studies which reported the highest assortativity in school going children [41] and young adults [1]. Age assortativity in mixing patterns is an important consideration for modelling disease outbreaks because higher assortativity values are linked to higher values of the reproductive ratio ($R_0$) of the infection [42]. After generating the ratio of the observed number of contacts to the values expected if mixing was random, contacts were found to be assortative (Q-index < 0.1), consistent with literature [19]. Contacts in summer were found to be less assortative compared to other seasons.

Epidemiological data on influenza cases in the Ballabgarh region could potentially be analysed with our results in mind. Chadha et al. [43] reported seasonal patterns in influenza cases in India, with recurring peaks in the monsoon and winter seasons in Delhi. This could be linked to the contact mixing patterns in these seasons, as well as factors like humidity and temperature, and could be examined in a further study. As our study investigates social contacts, our data can be used to parameterize age-stratified models for other communicable diseases besides influenza. Rohani et al. [44] illustrates that using age-structured contacts better replicates disease prevalence. Although our agent-based simulation showed marginal differences in the spread of COVID-19 across the three seasons in Ballabgarh, a wider study accounting for different geographic regions and communicable diseases could yield significant differences in their spread. Simulations making use of more detailed contact locations such as schools, public places, and hospitals should be carried out. This would allow the impact of different intervention strategies like age-specific vaccination and lockdown schemes to be investigated. With access to geocoded data, relations between the distances travelled by people and the number of contacts reported could also be studied.

## Study implications for public health modeling and interventions

Limiting social contacts of age-groups with a high number of contacts can prove beneficial in controlling an epidemic. The 0–9 and 10–19 year olds had the highest median contacts and the highest degree of centrality in the social network of Ballabgarh (S9 Fig). This age group reported several group contacts related to schools, madrasas, weddings, playing and worshipping. However, closing schools has other risks on the education system [45], and care should be taken in making this decision. Political events are accompanied by high numbers of group contacts and can exacerbate the spread of an infection. We observed that a majority of contacts outside the household are reported by middle-aged males, which could suggest that interventions targeted at middle-aged males could potentially help curtail the spread of a communicable disease. Weddings make up a very large number of group contacts, and should possibly be regulated in size if control of a communicable disease outbreak is prioritised by public health authorities. Contacts in transport and school settings have a very high proportion of physical contacts (in all three seasons), which may increase the probability of being infected at these locations. Wearing a mask and practicing proper hand hygiene can help to reduce the spread of infectious diseases. The winter season was found to have more contacts than the summer season, which in turn had more contacts than the monsoon season, which suggests that modifying the severity of an intervention depending on the season may be an effective policy. A staggered vaccination scheme may also prove beneficial. Most group contacts with large numbers of people took place outside of the home, which backs up the efficiency of lockdowns in epidemics. The demographic subgroup of 70+ year olds (at high risk of developing severe outcomes from influenza) had a high number of contacts with 10–19 and 20–29 year old categories (S3 Fig), which may help advise guidelines and policies to protect them.

## Limitations

Using contact diaries to keep track of contacts is suboptimal [46]. The number and duration of contact in contact diaries is often inaccurate due to recall bias, and the number of people in a group may be incorrectly reported due to human perception [47]. Respondents also tend to perceive time spent in some activities (talking, play, work) differently from reality [48] and tend to overestimate it [49,50]. Furthermore, a study [46] showed that people tend to forget contacts of short duration (< 5 mins) with a greater than 50% probability. We made every effort to reduce the burden on respondents by encouraging the reporting of individual and group contacts separately. This is in contrast to studies such as POLYMOD [1], where all contacts were reported as individual contacts. Reporting of group contacts allows to reduce the reporting bias of participants who encountered a large number of individuals. The data on duration of contacts is categorical, and for the analysis we assume durations are equally probable within a category by calculating the mean of each category. In practice, it is more common to have contacts of a shorter duration (i.e, more 4–6 hour contacts than 6–8 hour contacts) and thus the durations are potentially right-skewed.

As the survey was conducted among a convenience sample of households having a higher density of middle-aged (20–29) and old (60–69 years) individuals compared to the national statistics, the results are most representative of rural areas in Haryana, and may not generalize well to other areas. The surveys were conducted over three seasons, and as such there exists the possibility of reporting fatigue playing a role, as documented in other studies [46]. Furthermore, it is possible for each respondent to have a different reporting accuracy. The survey data was also gathered before the COVID-19 pandemic, which might have altered the contact patterns in present-day society. Finally, our conclusions are based on the changes observed in contact mixing patterns over seasons across one year, as opposed to a larger, long-term trend.

## Conclusion

Our findings show that seasonal variations exist in mixing patterns, and therefore disease transmission models should account for the temporal nature of social contacts. We examined contact mixing data for a town in rural India and provided inferences for policy-making. Further studies could explore the link between the contact patterns and communicable disease data.

## Supporting information

**S1 Fig. Respondent characteristics by season.** The age and gender stratified respondent counts per season are displayed in the grey blocks, while the blue and orange blocks show the loss and gain of respondents in follow-up surveys.
(TIF)

**S2 Fig. Percentage of group contacts in a season in each duration bucket.** Winter had a higher percentage of group contacts longer than 4 hours, compared to the other waves.
(TIF)

**S3 Fig. Number and durations of contacts by season.** Heatmaps representing mean numbers (red) and durations (blue, in person-hours) of contacts reported between age category dyads. Brighter colours imply a pair of categories with a high number/duration of contacts. Bright diagonal elements suggest some form of age-assortativity.
(TIF)

**S4 Fig. Number of contacts by season and gender.** Gender-stratified heat-maps representing the average number of contacts reported between two age categories (Males in blue, females in red). Brighter colours imply a pair of categories with a high number of contacts.
(TIF)

**S5 Fig. Duration of contacts by season and gender.** Gender-stratified matrices representing the average duration of contacts in person-hours reported between two age categories (Males in blue, females in red). Brighter colours imply a pair of categories with a high number of contacts.
(TIF)

**S6 Fig. Proportion of physical contacts.** (A) Barplots of the proportion of contacts reported to involve physical touch for all three waves, stratified by the duration of the contact. (B) Proportion of physical contacts for all three waves stratified by the frequency of which the respondent met the contact.
(TIF)

**S7 Fig. Gender stratified number of contacts.** (A) Boxplots of the number of contacts that occurred at home reported by males (blue) and females (red). (B) Boxplots of the number of total contacts reported to have occurred outside the home. Note that the upper y limit has been truncated to match that of (A). (C) Boxplots of the number of contacts stratified by gender and age category, across every wave.
(TIF)

**S8 Fig. Occupations and employment outside the home.** (A) Percentages of males and females classed as unemployed outside the home. (B) Distribution of total contacts for both employed and unemployed. Solid line represents a gaussian KDE. Note the longer tail on the distribution for employed respondents. (C) Frequencies of occupations among the respondents of the survey.
(TIF)

**S9 Fig. Social network of total contacts (individual and group) at Ballabgarh.** Directed graph representing the median number of contacts an age category has with every other age category. Higher numbers are depicted with stronger edge weights. Node sizes represent the total median contacts and node colours represent the PageRank of each node (darker = more important).
(TIF)

**S10 Fig. The disease progression model used in the agent-based simulation.** Figure sourced from Hazra et al.[34] S = susceptible, E = exposed, $I^p$ = Presymptomatic, $I^a$ = Asymptomatic, $I^m$ = Mildly Infected, $I^s$ = Severely Infected, H = Hospitalised, R = Recovered, D = Deceased.
(TIF)

**S11 Fig. Mean SIR curves from the simulation.** Blue, red and green represent susceptible, infectious, and removed counts respectively. Darker colours represent the winter, lighter colours monsoon and summer. Lines are the mean of 50 runs on a population of 10,000 for each season.
(TIF)

**S12 Fig. Age-stratified infectious curves from the simulation.** Mean number of infectious people across 50 simulations for every season who belong to a particular age category. Brown, red and orange represent winter, summer and monsoon seasons respectively. Significant

difference across seasons is observed in the 50–59 category.
(TIF)

**S13 Fig. SIR curves per season from the simulation.** Blue, red and green represent suscepti-
ble, infectious, and removed counts respectively. Each solid line is data from a single run,
dashed lines are the mean of 50 runs on a population of 10,000 for each season.
(TIF)

**S14 Fig. Distribution of number of contacts.** The number of individuals who reported a cer-
tain amount of total contacts is plotted against the number of total contacts. A curve of $y = \frac{y_0}{x^\alpha}$
is fit on the descending section of the data. Note that $\propto$ represents the negative slope of the
line on the log-log plot.
(TIF)

**S1 Table. Data cleaning and imputation methodology.**
(PDF)

**S2 Table. Keywords to map group contact reasons (free-write strings) to predefined cate-
gories.**
(PDF)

**S3 Table. Month and location wise break down of the number of people met in group con-
tacts.**
(PDF)

**S4 Table. Rates and parameters used in the agent-based simulation.** The rates used are
extracted from the Covasim model developed by Kerr et al.[33]
(PDF)

**S5 Table. Age stratified rates used in the simulation.** $r\_sus$ is relative susceptibility, and is
multiplied to the value of a target's *beta*. $p\_symp$ represents the probability of showing symp-
toms, $p\_severe$ the probability of displaying severe symptoms given the person is infected, and
$p\_deceased$ the probability of death given the person has severe symptoms. Rates taken from
the Covasim model developed by Kerr et al.[33]
(PDF)

## Author Contributions

**Conceptualization:** Sargun Nagpal, Rakesh Kumar, Supriya Kumar, Debayan Gupta,
Gautam I. Menon, Anand Krishnan.

**Data curation:** Sargun Nagpal, Rakesh Kumar, Riz Fernando Noronha, Supriya Kumar,
Debayan Gupta, Ritvik Amarchand, Mudita Gosain, Hanspria Sharma, Anand Krishnan.

**Formal analysis:** Sargun Nagpal, Rakesh Kumar, Riz Fernando Noronha, Supriya Kumar,
Debayan Gupta, Ritvik Amarchand, Mudita Gosain, Hanspria Sharma, Gautam I. Menon,
Anand Krishnan.

**Funding acquisition:** Supriya Kumar, Debayan Gupta, Gautam I. Menon, Anand Krishnan.

**Investigation:** Sargun Nagpal, Rakesh Kumar, Riz Fernando Noronha, Supriya Kumar,
Debayan Gupta, Ritvik Amarchand, Mudita Gosain, Hanspria Sharma, Gautam I. Menon,
Anand Krishnan.

**Methodology:** Sargun Nagpal, Rakesh Kumar, Riz Fernando Noronha, Supriya Kumar, Debayan Gupta, Ritvik Amarchand, Hanspria Sharma, Gautam I. Menon, Anand Krishnan.

**Project administration:** Rakesh Kumar, Supriya Kumar, Debayan Gupta, Gautam I. Menon, Anand Krishnan.

**Resources:** Rakesh Kumar, Supriya Kumar, Debayan Gupta, Gautam I. Menon, Anand Krishnan.

**Software:** Sargun Nagpal, Rakesh Kumar, Riz Fernando Noronha, Supriya Kumar, Debayan Gupta, Ritvik Amarchand, Anand Krishnan.

**Supervision:** Rakesh Kumar, Supriya Kumar, Debayan Gupta, Gautam I. Menon, Anand Krishnan.

**Validation:** Sargun Nagpal, Rakesh Kumar, Riz Fernando Noronha, Supriya Kumar, Debayan Gupta, Ritvik Amarchand, Gautam I. Menon, Anand Krishnan.

**Visualization:** Sargun Nagpal, Riz Fernando Noronha, Supriya Kumar, Debayan Gupta, Gautam I. Menon, Anand Krishnan.

**Writing – original draft:** Sargun Nagpal, Rakesh Kumar, Riz Fernando Noronha, Debayan Gupta, Ritvik Amarchand, Mudita Gosain, Hanspria Sharma, Gautam I. Menon.

**Writing – review & editing:** Sargun Nagpal, Rakesh Kumar, Riz Fernando Noronha, Supriya Kumar, Debayan Gupta, Gautam I. Menon, Anand Krishnan.

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
