## [Decision Letter · Decision Letter 0]

25 Jul 2023

PONE-D-23-06081Seasonal variations in social contact patterns in a rural population in north India: Implications for pandemic controlPLOS ONE

Dear Dr. Menon,

Thank you for submitting your manuscript to PLOS ONE. After careful consideration, we feel that it has merit but does not fully meet PLOS ONE’s publication criteria as it currently stands. Therefore, we invite you to submit a revised version of the manuscript that addresses the points raised during the review process.

We look forward to receiving your revised manuscript.

Kind regards,

Alberto Aleta

Academic Editor

PLOS ONE

“This work was supported by CDC co-operative agreement number 5U01IP000492 with AK and 14IPA141265003 with SK. Apart from the listed authors, the funders had no role in study design, data collection and analysis, decision to publish, or preparation of the manuscript.The findings and conclusions in this report are those of the authors and do not necessarily represent the official position of the Centers for Disease Control and Prevention”

Reviewers' comments:

Reviewer's Responses to Questions

**Comments to the Author**

1. Is the manuscript technically sound, and do the data support the conclusions?

Reviewer #1: Yes

Reviewer #2: Yes

2. Has the statistical analysis been performed appropriately and rigorously? 

Reviewer #1: Yes

Reviewer #2: Yes

3. Have the authors made all data underlying the findings in their manuscript fully available?

Reviewer #1: No

Reviewer #2: No

4. Is the manuscript presented in an intelligible fashion and written in standard English?

Reviewer #1: Yes

Reviewer #2: Yes

5. Review Comments to the Author

Reviewer #1: Summary. The authors provide a very interesting and needed addition to the literature. It is important to integrate heterogeneity in mixing patterns into inference and modeling of transmission dynamics, but typically these types of data are missing. The basic descriptions of the contact patterns presented by the authors alone are quite novel for South Asia. I only have a few minor questions and edits, which are listed below.

Methods, Data cleaning. The authors mentioned that, in part, information related to the nature and frequency of group contacts was used to impute gender, where missing. Can the authors comment on how this may impact their inference/conclusions related to differences social mixing characteristics by gender?

Methods, comparison of contacts … Can the authors clarify what is meant by “For this study, statistical methods for independent samples were used for comparison.”

Methods, Regression models to predict contacts and understand the effect of explanatory variables. Please provide more specificity with regard to how the negative binomial multivariate model(s) estimated the ‘independent effects’ of the characteristics listed. The reference to ‘hyperparameters’ suggests that you conducted these model fits from a Bayesian perspective. Is that correct, or am I reading too much into this statement?

Methods, Generating a synthetic population that mimics contact patterns. The authors refer to the use of a Markov Chain Monte-Carlo (MCMC) approach to generate synthetic population(s) from, in part, the assortative mixing matrices derived from their contact survey data. Can the authors provide more details about the specifics of this MCMC approach? Typically, MCMC refers to an approach for randomly sampling chains of inter-related events (specifically, where the probability of the current sample being drawn depends solely on the value of the preceding sample in the chain). This type of procedure doesn’t seem conducive to generating synthetic populations. Possibly the authors are referring to Monte-Carlo approaches to generating synthetic populations?

Results.

Since 95.4% of the respondents responded to surveys in more than one season, have the authors accounted for within-individual correlation in their analyses, for example, within the multivariable negative binomial models. Also, do the author’s hypothesis testing procedures for inter-season variability account for the very-high degree of participants responding in multiple seasons?

Figure 4. To aid in the interpretation of Figure 4, it might help to present the contact duration information on the hours-per-contact scale. It is hard to interpret and make use of information on the Total Duration scale, for example, to parameterize a stochastic individual-based simulator.

Figure S10. Legend. The authors state in the text that they used a individual-based stochastic epidemic simulator to simulate SARS-CoV-2/COVID-19 epidemics in their synthetic population, but this figure’s legend refers to a ‘compartmental model’ (i.e., a system of differential equations). Did the authors use a system of differential equations or a stochastic individual-based simulator with the possible states shown in Figure S10? If the latter, can the authors provide some explanation as to why the curves shown in Figure S11 and S12 seem to be overly smoothed for the typical outcome of a stochastic individual-based simulator?

In addition, 50 simulations seem to be a bit low number of simulation to run for the complex system represented by this synthetic population. Can the authors provide some plot or description of the epidemic curves from the simulated epidemics to provide the reader with some description of the variability in the resulting simulated epidemics.

Reviewer #2: The aim of this study was to explore seasonal variations in social contact patterns in a rural population in north India. This study analyses social contact data and compares differences in the number, duration and location of contacts by age group, gender, and season. Furthermore, the authors generate a social network using season-specific contact matrices to parameterize and simulate an agent-based model of COVID-19 transmission. The findings from this work can provide insights into the spread of infectious agents. This study is methodologically sound, and findings are appropriately discussed. The manuscript could be strengthened with a number of changes, for example improving clarity, further exploration or discussion of confounding factors and the context in which these findings can be applied.

Suggestions for improvement:

Introduction:

1. The introduction could benefit from reference to systematic reviews of contact patterns (both for high/low income countries), for example: Hoang et al, 2019 and Mousa et al, 2021.

2. The authors mention some of the studies which have attempted to measure and/or analyse temporal patterns. However, there is no summary of what were the main findings from these studies.

3. The authors could provide a very brief definition of empirical vs. synthetic contact matrices for the readers who are not familiar with the subject area.

Methods:

4. What are the implications of using a convenience sample in terms of generalizability? The authors could compare some of the characteristics of this convenience sample with population characteristics based on national statistics (e.g., age structure, gender etc.)

5. Was there a threshold number for defining “group contacts”? Initially, it was unclear whether the characteristics of each individual contact were recorded for group contacts. I suggest moving some of the information that is currently in the “age-assortative mixing matrices” earlier in the methods to improve clarity.

6. The duration of contacts is likely to be right-skewed. Taking the mean for each duration category assumes that the duration of contacts is uniformly distributed within each interval, which may not be the case. For example there may be a lot more contacts of duration 4-6 hours compared to 6-8 hours. If it is essential that a mean duration is computed, the authors should at least acknowledge the problem with this assumption.

7. The authors should justify the choice of a 5th degree polynomial. Does it have to do with the number of seasons examined, and does a higher degree polynomial run the risk of over-parameterizing?

Results:

8. Table 1: Columns should indicate whether the frequencies shown are the number of respondents or responses.

9. The directed graph is a useful way of visualizing the contact patterns in relation to age/gender. Is the thickness of the lines connecting the nodes proportional to the number of contacts between each pair of age-gender groups? If that is the case, it is surprising that the lines connecting the same age groups of different genders (e.g. 10-19M and 10-19F) have similar thickness to other pairs. Perhaps this could be adjusted to reflect level of contacts between groups?

10. Given that the authors have chosen to use non-parametric methods for comparing all outcomes, it may be preferable to always display boxplots with medians rather than barplots and means.

11. In general, the authors should include in all figure legends the total sample size making up each figure panel. E.g., for the seasonal comparisons, X respondents reported on Y contacts in at least 2 different seasons., etc.

12. P-values should be reported consistently to the nearest 2 (or 3) decimal places unless it is a very small value (then should be reported as p<0.001).

13. In the section “Number of contacts” the authors mention “….but were not significantly different from all other age categories”. This seems to contradict with what is shown in Fig 1D-1F (any comparison that is not dark blue seems to indicate a significant difference).

14. Loss to follow up seemed to be substantial after phase one (Figure S1). The authors may want to comment on whether those lost-to-follow-up differed in terms of age/gender/number of contacts.

15. “Middle-aged people (20-39)” , and “Middle aged males (20-49 years): I suggest avoiding such phrasing. You can refer to the age category directly e.g., “males aged …” or “participants aged …” .

Discussion

16. One of the main findings is that there were more contacts in the winter and less in the monsoon season, though explanations of this finding are not sufficiently discussed.

17. The authors should discuss whether the temporal effects are consistent spatially. How context-specific are these results in India, and do they have implications for other subnational or national areas?

18. The authors could make clear that the survey was conducted pre-COVID-19 pandemic, and that current contact patterns may be different.

19. School closures were not considered in the analysis, which is relevant for the most important nodes (age groups 10-19). This is likely to impact contact patterns, and it may be a good idea to include in the supplement an analysis (comparing “during school closures” vs. “during school term-time”) and/or in the discussion.

20. Have the authors considered looking at both age and gender assortativity? Only age assortativity is presented (and stratified by gender).

Supplementary material

21. The supplement could benefit from the addition of the questionnaire used for recording contacts.

22. The authors should cite all supplementary figures and tables in the main manuscript.

23. Please revise all supplementary figures as many of them contain spelling errors and errors in the axes names.

6. PLOS authors have the option to publish the peer review history of their article (what does this mean?). If published, this will include your full peer review and any attached files.

Reviewer #1: No

Reviewer #2: No

---

## [Author Response · Author response to Decision Letter 0]

28 Sep 2023

We attach a document detailing specific responses to all reviewer and editor comments

---

## [Decision Letter · Decision Letter 1]

7 Nov 2023

PONE-D-23-06081R1Seasonal variations in social contact patterns in a rural population in north India: Implications for pandemic controlPLOS ONE

Dear Dr. Menon,

Thank you for submitting your manuscript to PLOS ONE. After careful consideration, we feel that it has merit but does not fully meet PLOS ONE’s publication criteria as it currently stands. Therefore, we invite you to submit a revised version of the manuscript that addresses the points raised during the review process.

Request from PLOS Editorial Office: 

During the internal evaluation of your manuscript, we noted that the ethics section states: “We received written consent from all participants over 7 years of age. Caregivers provided written consent for participants below 7 years of age.” According to the ICMR guidelines (https://main.icmr.nic.in/sites/default/files/guidelines/National_Ethical_Guidelines_for_BioMedical_Research_Involving_Children_0.pdf), we would expect written informed consent from parents/LARs for any participant aged 7 to 18, in addition to the assent provided by the child participants. Please update your methods section and ensure you have also stated whether you obtained consent from parents or guardians of the minors included in the study or whether the research ethics committee or IRB specifically waived the need for their consent.

Please note that the addition of this information is essential for your manuscript to be considered for publication. Thank you very much for your attention to this request.

We look forward to receiving your revised manuscript.

Kind regards,

Johanna Pruller, PhD

Associate Editor

PLOS ONE

on behalf of

Alberto Aleta

Academic Editor

PLOS ONE

Journal Requirements:

Additional Editor Comments (if provided):

Reviewers' comments:

Reviewer's Responses to Questions

**Comments to the Author**

1. If the authors have adequately addressed your comments raised in a previous round of review and you feel that this manuscript is now acceptable for publication, you may indicate that here to bypass the “Comments to the Author” section, enter your conflict of interest statement in the “Confidential to Editor” section, and submit your "Accept" recommendation.

Reviewer #1: All comments have been addressed

Reviewer #2: All comments have been addressed

2. Is the manuscript technically sound, and do the data support the conclusions?

Reviewer #1: Yes

Reviewer #2: Yes

3. Has the statistical analysis been performed appropriately and rigorously? 

Reviewer #1: Yes

Reviewer #2: Yes

4. Have the authors made all data underlying the findings in their manuscript fully available?

Reviewer #1: Yes

Reviewer #2: Yes

5. Is the manuscript presented in an intelligible fashion and written in standard English?

Reviewer #1: Yes

Reviewer #2: Yes

6. Review Comments to the Author

Reviewer #1: (No Response)

Reviewer #2: The authors have sufficiently addressed my previous comments and suggestions, and I have nothing further to add.

7. PLOS authors have the option to publish the peer review history of their article (what does this mean?). If published, this will include your full peer review and any attached files.

Reviewer #1: No

Reviewer #2: No

---

## [Author Response · Author response to Decision Letter 1]

8 Nov 2023

We thank the Editor for bringing an incomplete ethics statement to our attention.

We have now changed the lines:

 “We received written consent from all participants over 7 years of age. Caregivers provided written consent for participants below 7 years of age.”

To

“We received written consent from all participants 18 years of age and above. All those between 7 years and 18 years provided written assent in addition to the written consent of the caregivers. Caregivers provided written consent for participants below 7 years of age.”

This reflects what was actually done. The earlier version was incomplete.

---

## [Editor Report · Decision Letter 2]

14 Dec 2023

Seasonal variations in social contact patterns in a rural population in north India: Implications for pandemic control

PONE-D-23-06081R2

Dear Dr. Menon,

We’re pleased to inform you that your manuscript has been judged scientifically suitable for publication and will be formally accepted for publication once it meets all outstanding technical requirements.

Kind regards,

Alberto Aleta

Academic Editor

PLOS ONE
---

## [Editor Report · Acceptance letter]

10 Feb 2024

PONE-D-23-06081R2 

PLOS ONE

Dear Dr. Menon, 

I'm pleased to inform you that your manuscript has been deemed suitable for publication in PLOS ONE. Congratulations! Your manuscript is now being handed over to our production team.

Kind regards, 

on behalf of

Dr. Alberto Aleta 

Academic Editor

PLOS ONE